# Development of E-ice-COLD-PCR assay combined with HRM analysis for *Nucleophosmin1* gene mutation detection in acute myelogenous leukemia

Rattana Kongta[1], Noppamas Panyasit[2], Wuttichote Jansaento[3], Suwit Duangmano[1,4]*

1 Department of Medical Technology, Faculty of Associated Medical Sciences, Chiang Mai University, Chiang Mai, Thailand, 2 Division of Hematology, Department of Internal Medicine, Faculty of Medicine, Chiang Mai University, Chiang Mai, Thailand, 3 Faculty of Medical Technology, Nation University, Lampang, Thailand, 4 Cancer Research Unit of Associated Medical Sciences (AMS-CRU), Faculty of Associated Medical Sciences, Chiang Mai University, Chiang Mai, Thailand

* suwit.du@cmu.ac.th

**Data Availability Statement:** All relevant data are within the paper and its Supporting information files.

## Abstract

Mutations of the n*ucleophosmin1* (*NPM1*) gene represent the most frequent molecular alteration in acute myelogenous leukemia (AML), especially in patients with AML who have a normal karyotype. These alterations have been shown to carry favorable prognostic significance in patients with AML. Several methods have been developed for detection of *NPM1* gene mutations. However, their ability to detect low levels of mutations in a wild-type background is limited. In this study, the Enhance improved and complete enrichment Co-amplification at Lower Denaturation temperature Polymerase Chain Reaction (E-ice-COLD-PCR) assay combined with High Resolution Melting (HRM) analysis was developed and validated for highly specific and sensitive screening for *NPM1* gene mutations. A total of 83 blood samples from patients with AML were collected, and their DNA was extracted. For mutational analysis, the E-ice-COLD-PCR assay for the detection of *NPM1* gene mutations was developed. PCR products were analyzed by HRM analysis. All positive samples were confirmed by direct sequencing. This assay enabled detection specificity and sensitivity of *NPM1* mutations in 9/83 patients with AML. Direct sequencing results were 100% concordant with this method. In addition, the limit of detection was 12.5% mutant in the final concentration of 5 ng genomic DNA. The E-ice-COLD-PCR assay with HRM analysis is a highly specific and sensitive screening method for enrichment of detecting *NPM1* gene mutations. This method has both a short turn around time and easier interpretation compared to those of other methods.

## Introduction

Acute myelogenous leukemia (AML), a hematological malignancy with a high mortality rate, is characterized by the uncontrolled proliferation of white blood cells and the loss of their

**Funding:** We are grateful for research fund from National Research Council of Thailand (NRCT), grant no. GSCMU (NRCT)/15/2563 to RK, Health System Research Institute (HSRI), grant no. HSRI 63-079 to SD, Chiang Mai University Grant for New Researcher to SD.

**Competing interests:** The authors have declared that no competing interests exist.

ability to normally differentiate. AML is the highest incidence rate among hematological malignancies in Thailand [1]. There are several factors that support the development of AML, including chronic or high-dose chemical exposure; high levels of radiation exposure; receipt of chemotherapy in previous cancer treatment; and genetic disorders, such as chromosomal abnormalities and gene mutations. Among these factors, gene mutations are the most common cause of AML. Recently, researchers have suggested that mutations of the *nucleophosmin1* (*NPM1*) gene represent the most frequent molecular alterations in AML that affect the processes of cellular differentiation and apoptosis, especially in the presence of a normal karyotype [2, 3]. In Thailand, the prevalence of *NPM1* mutations in patients with AML were 38.1% [4] and 17.5% [5] in central Thailand and in upper northern Thailand, respectively. *NPM1* gene mutations are characteristically heterozygous and are mainly insertions of nucleotide clusters within exons 11 [6] and 12 [7] of the *NPM1* gene. The insertions are typically tetranucleotide insertions at positions 956 to 971. The most prevalent mutation is insertion of TCTG in exon 12 of the *NPM1* gene or type A mutation, leading to a frameshift mutation and abnormally elongated proteins, resulting in abnormalities of the functions involved in ribosome biogenesis, centrosome duplication, genomic stability, controlling DNA repair, cell proliferation, and apoptosis. Increasing evidence of *NPM1* gene mutations has been found in AML, and AML with *NPM1* gene mutations has been separated as a provisional entity for the 2016 World Health Organization (WHO) classification of AML [8]. Additionally, these alterations have been shown to carry favorable prognostic significance in patients with AML because of their clinical importance in terms of risk assessment and treatment decisions. Therefore, the analysis of *NPM1* gene mutational status should be usable after the diagnosis.

To date, several PCR-based methods have been developed for detection of *NPM1* mutations, including PCR amplification and direct sequencing [9, 10], high-resolution fragment analysis [11, 12], melting curve analysis, denaturing high-performance liquid chromatography (DHPLC) [4, 13], locked nucleic acid (LNA) PCR assay [14–16], allele-specific oligonucleotide (ASO) PCR [17], and FAST PCR assay with agarose E-gel electrophoresis [18]. However, most of these methods are technically challenging, complicated, and expensive. Most importantly, they do not have the ability to detect low levels of mutant alleles in a wild-type background. Enhance improved and complete enrichment co-amplification at lower denaturation temperature polymerase chain reaction (E-ice-COLD-PCR) is a modified PCR-based method that is usable for the enrichment of all types of mutations. This method is based on a non-extendable blocker probe that is incorporated with chemically modified nucleotides or LNA bases into the blocker probe. The addition of LNA bases into the blocker probe increases the stability and the melting temperature (Tm) of the complete match DNA-LNA probe heteroduplex, leading to a high Tm difference between the complete match and mismatch DNA-LNA heteroduplex. The blocker probe overlaps six bases at the 3' end of only one primer, which could be hybridized by the complete match with the wild-type sequence while being an incomplete match with the mutant sequence. After the denaturation step in each cycle, a blocker probe hybridizes to the target sequence at 70˚C and the blocker probe allows for the complete denaturation of the mutant-LNA probe heteroduplex at the critical temperature (Tc), while the WT-LNA probe heteroduplex remains strongly hybridized, which only allows the amplification of the mutant. The E-ice-COLD-PCR assay is the strongest for the detection of mutations compared with full-, fast- and ice-COLD-PCR methods [19].

Mutation scanning with high-resolution melting (HRM) analysis consists of post-COLD-PCR analysis with a continuous increase in temperature from 65˚C to 95˚C to identify the mutations based on detecting small differences in PCR melting (dissociation) curves. The signal differences of PCR products are generated from the transition of double stranded DNA (dsDNA) to single stranded DNA (ssDNA) in the presence of the fluorescent intercalating dye.

The fluorescent dye is released when dsDNA denatures into ssDNA, resulting in the loss or decrease of the fluorescent signal at the Tm of the PCR products. In addition, HRM is a rapid and inexpensive scanning method and can be performed without opening the PCR tubes, which prevents contamination [20]. Therefore, the combination of COLD-PCR with HRM analysis could improve the mutation scanning ability of COLD-PCR.

Significantly, the advent of personalized medicine exhibits a paradigm shift in the treatment of AML, in which both the genetic and molecular analyses of patients with AML are more important than both morphological and cytochemical testing to guide the treatment and response of the patient. The evaluation of *NPM1* gene mutations is of great importance for risk stratification in the patient and treatment decisions of the physician, resulting in decreasing mortality rates from AML. Moreover, AML with *NPM1* gene mutations is a separate entity in the revised 2016 World Health Organization (WHO) classification. Therefore, establishing sensitive and reliable E-ice-COLD-PCR methods with HRM analysis for detecting lower levels of *NPM1* gene mutations in a wild-type background is essential for individualized patient management.

## Materials and methods

### Blood sample collection

EDTA blood samples were all leftover from 83 patients diagnosed with AML at Maharaj Nakorn Chiang Mai Hospital, Chiang Mai, Thailand. The use of de-identified blood samples was approved by the Ethics Committee of the Faculty of Associated Medical Sciences, Chiang Mai University, Chiang Mai, Thailand (exempted number AMSEC-62EM-018) and Research Ethics Committee, Faculty of Medicine, Chiang Mai University, Chiang Mai, Thailand (exempted number EXEMPTION-6485/2019). The sample size was calculated using the finite population proportion formula by free application (n4Studies). The DNA extraction was performed within 24 hours of obtaining blood samples using the NucleoSpin® Blood DNA extraction kit (MACHEREY-NAGEL GmbH & Co. KG, Germany) according to manufacturer's instructions. The extracted DNA was quantified and assessed for purity by a microplate spectrophotometer and stored at -20˚C.

### Locked nucleic acid (LNA) probe designed

The LNA probe incorporated one LNA base into the blocker probe. Its design was dependent on the primer and overlapped six bases at the 3' end of the forward primer, which could be hybridized with the complete match with the *NPM1* wild-type template while the incomplete match was with the mutant template. The presence of a LNA base in the blocker probe allows for the complete denaturation of the mutant-LNA probe heteroduplex at the Tc, while the WT-LNA probe heteroduplex remained strongly hybridized, which only allowed for the amplification of mutants. The 3' end of the LNA probe was modified by phosphorylation to prevent the extension by the DNA polymerase enzyme.

### Detecting *NPM1* gene mutations in patient samples using a standard PCR assay

The standard PCR assay was performed on 83 patient samples to detect *NPM1* mutations. The PCR reaction in a final volume 20 μL contained 10 μL of 2X Quick Taq HS DyeMix (Toyobo, Japan), 0.2 μM of forward primer (NPM-F:5'-GTGTTGTGGTTCCTTAACCACAT-3') and reverse primer (5'-CTGTTACAG AAATGAAATAAGACGGAAA-3'), approximately 100 ng of patient DNA, and ddH20 (up to 20 μL). KG-1a DNA and synthesized mutated-*NPM1*

ssDNA fragments were used as the wild-type control and mutated control, respectively. PCR was performed in a Mini MJ Thermal Cycler (Bio-rad, Inc., Singapore), including initial denaturation 95˚C for 3 min, followed by 35 cycles of 95˚C for 30 s, 60˚C for 30 s and 72˚C for 30 s, with a final extension at 72˚C for 3 min. PCR products were obtained on 10% polyacrylamide gel electrophoresis using vertical gel electrophoresis, run at 400 volts for 150 min and visualized by a molecular imager to determine the amplicon of both wild-type and mutant products. All PCR products were confirmed by direct sequencing in both directions using fluorescent dye-terminator sequencing on ABI 3730xl DNA sequencers (Applied Biosystems™, Thermo Fisher Scientific Inc. MA, USA).

## Development of the E-ice-COLD-PCR assay

The feature of the E-ice-COLD-PCR assay is the Tc for selectively denaturing mutant-LNA probe heteroduplexes, while the WT-LNA probe heteroduplexes remain strongly hybridized to allow enhancement of the mutant alleles of *NPM1* mutations. To determine the Tc of the E-ice-COLD-PCR assay, each PCR reaction in final volume of 20 μL contained 10 μL of 2x Sensi-FAST HRM Master Mix (Meridian Bioscience, Inc., USA), 0.2 μM of forward primer (NPM-F:5'-GTGTTG TGGTTCCTTAACCACAT-3') and reverse primer (5'-CTGTTA CAGAAATGAAATA AGACGGAAA-3'), 400 nM of LNA probe (NPM-LNA Probe: 5'-CCACATTTCTTTTTTTTTTTTCC AGGCTATTC AAGATCTCTG+GC-Ph-3'), approximately 5 ng of KG-1a DNA or 1 ag of synthesized mutated *NPM1* ssDNA fragments and ddH$_2$0 (up to 20 μL). PCR was performed in the CFX96 Touch Real-Time PCR System (Bio-Rad laboratories, Inc., USA), including initial denaturation at 95˚C for 3 min; followed by three cycles of 95˚C for 5 s, 60˚C for 10 s and 72˚C for 30 s; then followed by 40 cycles of denaturation at 95˚C for 5 s, LNA probe hybridization at 70˚C for 30 s, various Tc ranges (80–90˚C) for 40 s, primer annealing at 60˚C for 10 s and extension at 72˚C for 30 s, with a final extension at 72˚C for 3 min. The final step was a melting curve with a continuous increase in temperature from 65˚C to 85˚C (0.2˚C per acquisition and 5 s hold before each acquisition). Finally, HRM was analyzed using Precision Melt Analysis™ software for the detection of mutations.

After determination of the Tc, the E-ice-COLD-PCR assay with HRM analysis were used to verify the enrichment of the detection of *NPM1* mutations in four types of mutations, including *NPM1* mutation types A, B, D and J. Synthesized ssDNA fragments of the four types of *NPM1* mutations were used as the mutated template, and KG-1a DNA was used as the wild-type template. In brief, the PCR reaction in a final volume of 20 μL contained 10 μL of 2x Sen-siFAST HRM Master Mix (Meridian Bioscience, Inc., USA), 0.2 μM of each primer, 400 nM of LNA probe, approximately 5 ng of KG-1a DNA or 1 ag of each synthesized mutated-*NPM1* ssDNA fragments and ddH$_2$0 (up to 20 μL). PCR was performed in the CFX96 Touch Real-Time PCR System (Bio-Rad laboratories, Inc., USA), including initial denaturation at 95˚C for 3 min; followed by three cycles of 95˚C for 5 s, 60˚C for 10 s and 72˚C for 30 s; then followed by 40 cycles of denaturation at 95˚C for 5 s, LNA probe hybridization at 70˚C for 30 s, critical denaturation at 88˚C for 40 s, primer annealing at 60˚C for 10 s and extension at 72˚C for 30 s, with a final extension at 72˚C for 3 min. The final step was a melting curve with a continuous increase in temperature from 65˚C to 85˚C (0.2˚C per acquisition and 5 s hold before each acquisition), and HRM was analyzed using Precision Melt Analysis™ software to determine the sensitivity of the E-ice-COLD-PCR assay.

## LOD of the E-ice-COLD-PCR assay

The mutated *NPM1* template was diluted in the *NPM1* wild-type template. All dilutions were performed by the E-ice-COLD-PCR assay combined with HRM analysis. In brief, each PCR reaction in a final volume of 20 μL contained 10 μL of 2x SensiFAST HRM Master Mix (Meridian Bioscience, Inc., USA), 0.2 μM of each forward primer, 400 nM of LNA probe, approximately 5 ng of DNA and $ddH_2O$ (up to 20 μL). PCR was performed in the CFX96 Touch Real-Time PCR System (Bio-Rad laboratories, Inc., USA), including initial denaturation at 95˚C for 3 min, three cycles of 95˚C for 5 s, 60˚C for 10 s and 72˚C for 30 s, followed by 40 cycles of denaturation at 95˚C for 5 s, LNA probe hybridization at 70˚C for 30 s, critical denaturation at 88˚C for 40 s, primer annealing at 60˚C for 10 s and extension at 72˚C for 30 s, with a final extension at 72˚C for 3 min. The final step was a melting curve analysis with a continuous increase in temperature from 65˚C to 85˚C (0.2˚C per acquisition and 5 s hold before each acquisition), and HRM was analyzed using Precision Melt Analysis™ software to determine the LOD of the E-ice-COLD-PCR assay.

## Detecting *NPM1* gene mutations in patient samples using the E-ice-COLD-PCR assay

The E-ice-COLD- PCR assay was performed on 83 patient samples to detect *NPM1* mutations. In brief, each PCR reaction in a final volume of 20 μL contained 10 μL of 2x SensiFAST HRM Master Mix (Meridian Bioscience, Inc., USA), 0.2 μM of each primer, 400 nM of LNA probe, approximately 5 ng of patient DNA and $ddH_2O$ (up to 20 μL). KG-1a DNA and synthesized mutated-*NPM1* ssDNA fragments were used as the wild-type control and mutated control, respectively. PCR was performed in the CFX96 Touch Real-Time PCR System (Bio-Rad laboratories, Inc., USA), including initial denaturation at 95˚C for 3 min, three cycles of 95˚C for 5 s, 60˚C for 10 s and 72˚C for 30 s, followed by 40 cycles of denaturation at 95˚C for 5 s, LNA probe hybridization at 70˚C for 30 s, critical denaturation at 88˚C for 40 s, primer annealing at 60˚C for 10 s and extension at 72˚C for 30 s, with a final extension at 72˚C for 3 min. The final step was a melting curve analysis with a continuous increase in temperature from 65˚C to 85˚C (0.2˚C per acquisition and 5 s hold before each acquisition), and HRM was analyzed using Precision Melt Analysis™ software for detecting *NPM1* mutations in patient samples. Positive samples from the E-ice-COLD-PCR assay combined with HRM analysis were confirmed by direct sequencing in both directions using fluorescent dye-terminator sequencing on ABI 3730xl DNA sequencers (Applied Biosystems™, Thermo Fisher Scientific Inc. MA, USA).

## Statistical analysis

The positive predictive value (PPV), negative predictive value (NPV), sensitivity, and specificity of the E-ice-COLD-PCR assay were determined using direct sequencing as the reference method. The PPV was defined as [the number of true positives / (the number of true positives + the number of false positives)] *100. The NPV was defined as [the number of true negatives / (the number of true negatives+ the number of false negatives)] *100. The sensitivity was defined as [the number of true positives / (the number of true positives+ the number of false negatives)] *100. The specificity was defined as [the number of true negatives / (the number of true negatives+ the number of false positives)] *100.

## Results

### Detecting *NPM1* gene mutation in patient samples using standard PCR assay

A total of 83 patient samples were included in this study—nine samples contained *NPM1* gene mutations and 74 samples were wild-type as determined by standard PCR assay followed by 10% polyacrylamide gel electrophoresis. The mutant showed a banding pattern size of 144 bp, while the wild-type showed a banding pattern size of 140 bp. The mutated PCR products from patient samples showed two distinct banding patterns on polyacrylamide gel (Fig 1 and S1 Fig). All PCR products were confirmed by direct sequencing in both directions. The results exhibited that eight samples (including sample no. 3, 41, 44, 47, 50, 57, 62 and 68, respectively) were *NPM1* mutation type A, whereas 75 samples were wild-type. One sample (sample no. 11) showed inconsistent results with the standard PCR assay followed by polyacrylamide gel electrophoresis and direct sequencing.

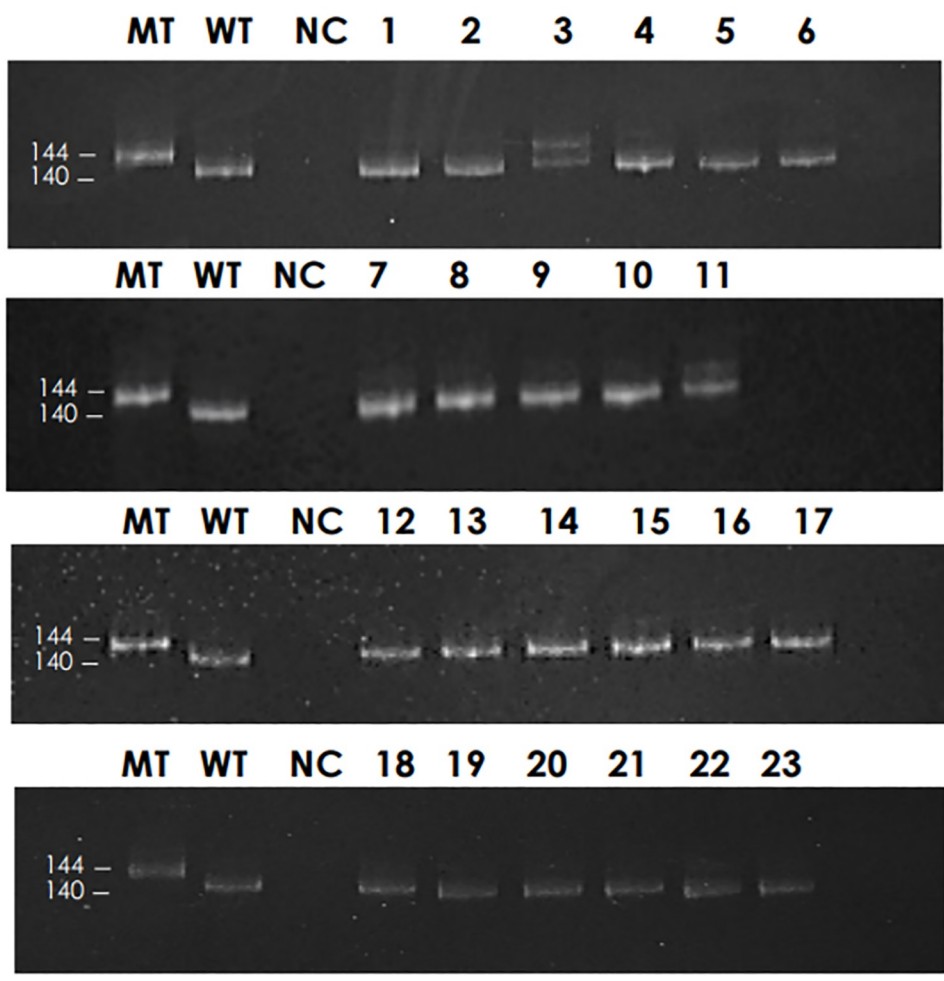

**Fig 1. Detecting *NPM1* gene mutations in patient samples using a standard PCR assay.** The PCR products are visualized on 10% polyacrylamide gel. MT indicates *NPM1* mutated control and WT indicates *NPM1* wild-type control. A double band indicates a heterogeneous mutation.

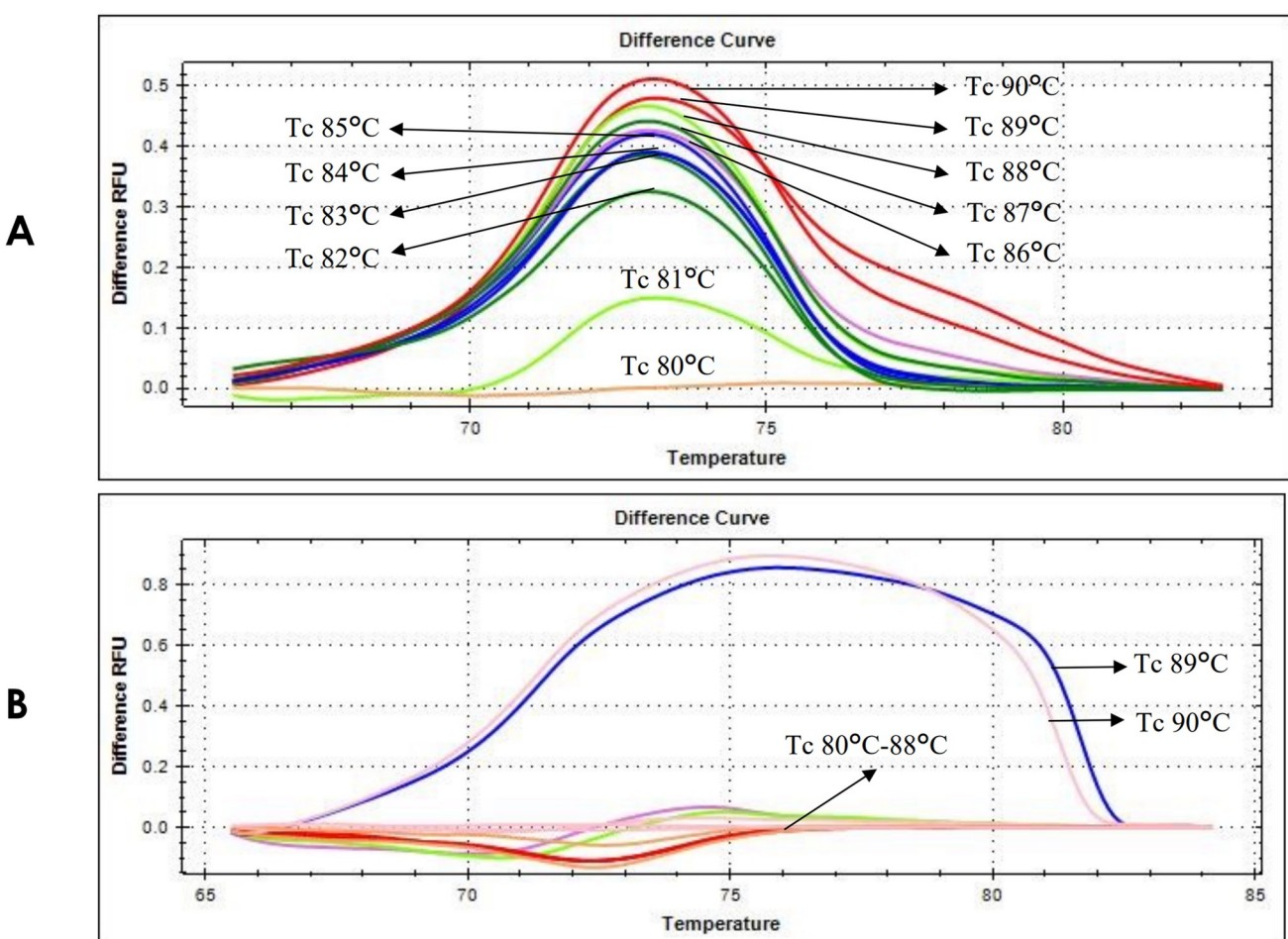

**Fig 2. HRM difference curves of Tc determinations.** (a) The height of the difference curve for mutants increased with increasing Tc from 80–90˚C. (b) No signals of wild-type products were produced when increasing the Tc from 80–88˚C, while the wild-type was amplified when increasing the Tc to more than 88˚C and presented the signal in the HRM difference curve.

## Tc of E-ice-COLD-PCR assay

The Tc was determined to select the best Tc with the highest enrichment of the mutant product amplification, while the wild-type products could not be produced. The determination of the Tc was performed in *NPM1* wild-type and *NPM1* mutation type A as representations of *NPM1* mutations. The E-ice-COLD-PCR assay had varying levels of Tc, range 80–90˚C. PCR products were analyzed by HRM analysis, generating the data into HRM difference curves. The results showed that the Tc at 80–88˚C was able to selectively denature and amplify only mutant products (Fig 2a), while the wild-type product could not be produced (Fig 2b). The enrichment of mutants increased from the Tc 80–88˚C, which was represented by the increasing height of the peak. Whereas, the Tc at 89–90˚C highly enhanced the amplification of mutants. However, it could not selectively amplify only mutant products, because the wild-type product could also be amplified (Fig 2b). Therefore, the Tc of 88˚C was selected for the highest enrichment of the detection of *NPM1* mutations in the E-ice-COLD-PCR assay.

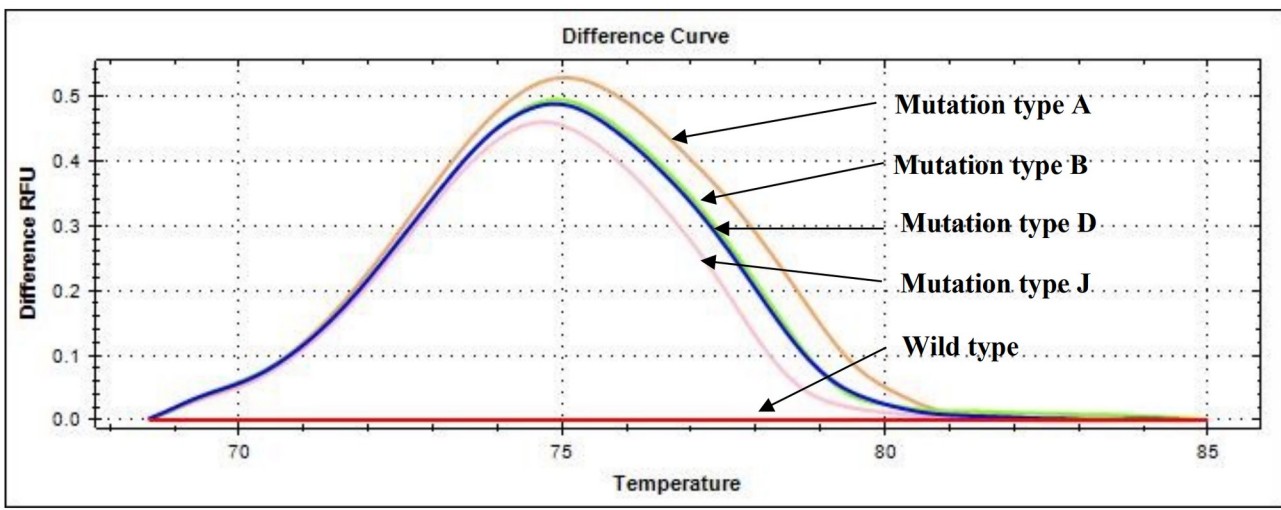

**Fig 3. The E-ice-COLD-PCR assay combined with HRM analysis of four types of *NPM1* mutations.** Following the E-ice-COLD-PCR assay, the PCR products were analyzed by HRM analysis. The HRM difference curves show the peaks of *NPM1* mutation types A, B, D, and J, while no peak in the *NPM1* wild-type indicated that the E-ice-COLD-PCR assay can inhibit wild-type amplification. Meanwhile, it can enhance the amplification of the four types *NPM1* mutations.

### Verification of E-ice-COLD-PCR for detection of *NPM1* mutations

The E-ice-COLD-PCR assay was performed in four types of *NPM1* mutations, including *NPM1* mutation types A, B, D, and J to verify the E-ice-COLD-PCR method for detection of *NPM1* mutations covering the four mutation types. Following the E-ice-COLD-PCR assay, the PCR products were analyzed by HRM analysis, and the PCR products were also detected by agarose gel electrophoresis. The results showed that the E-ice-COLD-PCR assay was able to amplify the mutant PCR products in all four types of *NPM1* mutations, while the wild-type product could not be produced (Fig 3).

### Limit of detection of the E-ice-COLD-PCR assay

The limit of detection (LOD) of the E-ice-COLD-PCR assay was the lowest concentration of the DNA sample that can reliably detect NPM1 gene mutations. In this study, the mutant DNA was diluted in the wild-type DNA for various final concentrations (100%, 50%, 25%, 12.5%, 6.25%, 3.13%, and 0%). A total of 5 ng DNA was amplified as a template in each PCR reaction. The PCR products were analyzed by HRM analysis. The results showed that the peak or signal of different relative fluorescence units on the HRM difference curves decreased from the decreasing final concentration of mutant DNA 100% to 12.5%, and the peak or signal of different relative fluorescence units disappeared at the decreasing final concentration of mutant DNA from 6.25% to 0% (Fig 4). The lowest amount of mutant DNA detecting *NPM1* mutations was 12.5% of the mutants in the final concentration of 5 ng DNA. Therefore, the LOD of the E-ice-COLD-PCR assay was 12.5% of the mutants in the final concentration of 5 ng DNA.

### Detecting *NPM1* gene mutations in patient samples using the E-ice-COLD-PCR assay

Detecting *NPM1* mutations were performed in a total of 83 patient samples using the E-ice-COLD-PCR assay. The PCR products were analyzed by HRM analysis, and the data generated

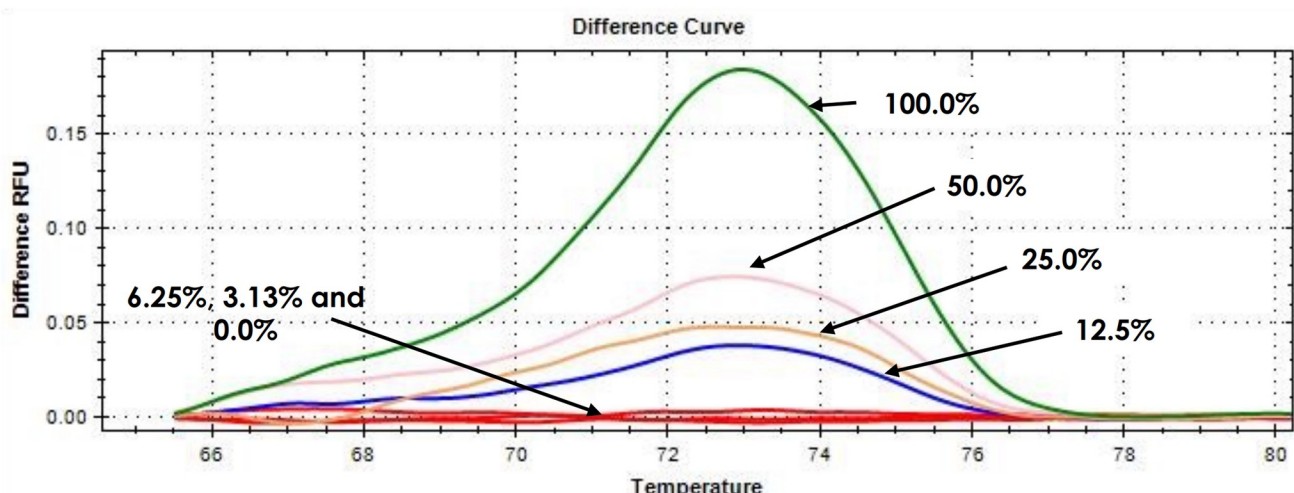

**Fig 4. LOD of the E-ice-COLD-PCR assay combined with HRM analysis.** Following the E-ice-COLD-PCR assay, the PCR products were analyzed by HRM analysis. The HRM difference curves show the height of the curves decreased with the decreasing percentage of mutants, and the peak disappeared with the decreasing percentage of mutants from 6.25% to 0%.

for each sample as the HRM difference curves in Fig 5 and S2 Fig. The results from 83 samples showed that nine samples contained *NPM1* mutations, and the remaining samples were wild-type. All PCR products were confirmed by direct sequencing in both directions. The results were in 100% concordance with the E-ice-COLD-PCR assay. The most common mutation was type A, which is caused by insertion of TCTG (Fig 6b). The sensitivity and specificity of the E-ice-COLD-PCR assay for the detection of *NPM1* mutations were both 100%. All positive samples carried mutations, and none of the negative samples contained mutations. The positive predictive value (PPV) was 100%, and the negative predictive value (NPV) was 100%.

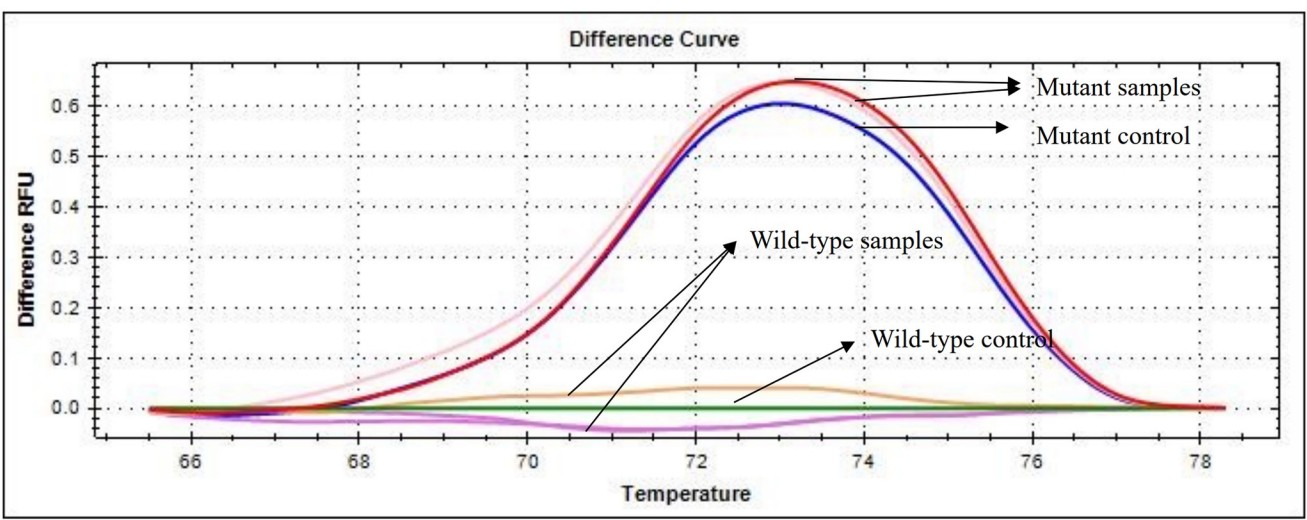

**Fig 5. Detecting *NPM1* gene mutations by HRM analysis.** Detecting *NPM1* gene mutations using the E-ice-COLD-PCR assay and generating the data into HRM difference curves. Red line indicates mutant control. Green line indicates wild-type control. The wild-type samples present the peak at the same location as the wild-type control, and the mutant samples present the peak at the same location as the mutant control.

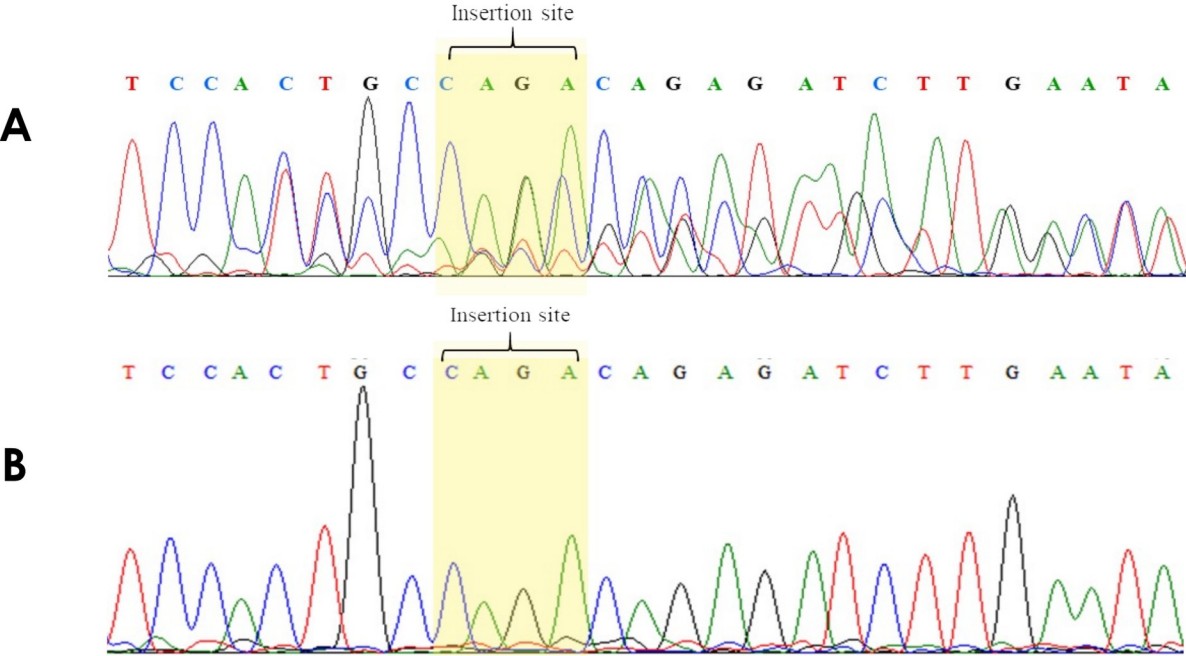

**Fig 6. Direct sequencing in the reverse direction of mutated products.** (a) Mutated products from the standard PCR assay (b) Mutated products from the E-ice-COLD-PCR assay. The results showed that the *NPM1* mutation type A is a 4 bp insertion.

## Discussion

AML, a hematological disorder characterized by the uncontrolled proliferation of white blood cells and reduced hematopoiesis, is the highest incidence rate of hematological malignancies in Thailand. Several factors have been reported to support the development of AML, most of which are genetic alterations or gene mutations. Many reports have found that the *nucleophosmin1* (*NPM1*) gene represents the most frequent molecular alteration in AML, especially in the presence of a normal karyotype. *NPM1* gene mutations occur in approximately one-third of patients with AML. In Thailand, the most prevalent *NPM1* mutation is the type A mutation, which inserts TGTC into exon 12 of the *NPM1* gene, followed by type B, type D, and type J mutations that insert CATG, CCTG, and TATG, respectively [4]. *NPM1* mutations resulted in elongated proteins and abnormalities of their functions involved in ribosome biogenesis, centrosome duplication during mitosis, genomic stability, controlling DNA repair, and the interactions with the oncosuppressors p53 and ARF and their partners, such as Hdm2/Mdm2, for controlling cell proliferation and apoptosis. *NPM1* mutations carry a favorable prognostic significance in patients with AML because of their clinical importance in terms of risk assessment and treatment decisions. Patients with AML who carried *NPM1* gene mutations in the absence of *FLT3*-ITD mutations were classified into a favorable prognosis group [21] that is susceptible to chemotherapy as compared to patients with AML without *NPM1* mutations and showed the best complete remission rate among these patients. The *NPM1* protein normally translocates between the nucleus and cytoplasm. It is involved in protecting DNA polymerase eta in the nucleus, controlling DNA repair via the translesion synthesis (TLS) pathway by binding to DNA polymerase eta for prevention of proteasome degradation, while mutant NPM1 protein accumulated in the cytoplasm, resulting in increased DNA polymerase eta degradation and failure to protect the cell from chemotherapy-induced DNA lesions via the TLS pathway [22].

It has been indicated that the addition of chemotherapy is enough for the treatment of patients with AML with *NPM1* mutations in the absence of *FLT3*-ITD mutations, and it is unnecessary to perform autologous or allogenic stem cell transplantation. In addition, the most commonly co-occurring mutation with *NPM1* in AML is cohesion mutation. When compared their variant allele frequency (VAF) relationships to survival, If *NPM1* mutations occurred before a chromatin/cohesin variant, there are association with a good response to treatment and five-year survival [23]. The variant of *NPM1* is also importance for AML prognosis. Therefore, after the diagnosis, the detection of *NPM1* gene mutation status should be usable for patient prognosis and making decisions about treatment.

Several methods have been developed for detecting *NPM1* mutations, such as PCR amplification and direct sequencing, high-resolution fragment analysis, PCR with capillary electrophoresis, melting curve analysis, DHPLC, LNA-mediated PCR, ASO-PCR and FAST PCR assays with agarose E-gel electrophoresis, but most of these methods are technically challenging, complicated, expensive, and require additional laboratory equipment that is not routinely available in some molecular laboratories. Most importantly, the inability to detect low levels of mutations in a wild-type background. Because these methods are standard PCR-based methods. They could be amplified the amplicon not only mutant but also wild-type amplicon. After PCR amplification, the wild-type PCR products were generated more than the mutant PCR products in case of low levels of mutations in a wild-type background that affects their result interpretation in post-PCR analysis. For example, the PCR products were analyzed by agarose E-gel or non- saturated acrylamide gel electrophoresis and their represented the vary faint band of mutant on the gel resulting in misdetection of *NPM1* mutations as well as standard sequencing, the signal of wild-type amplicons were overlaid the signal of mutant amplicons in standard sequencing. Even detecting *NPM1* mutations by melting curve analysis or DHPLC, the interpretation of *NPM1* mutations were also difficult because the signal of mutant was overlaid by wild-type. Finally, the misdetection of *NPM1* mutations lead to decision-making mistake in AML patient treatment. To enrich low level *NPM1* mutation detection, the E-ice-COLD-PCR assay is a new PCR-based method used for enrichment of low levels of mutated genes in mixtures of wild-type and mutant alleles by blocked amplification of wild-type. Several studies have reported that the E-ice-COLD-PCR assay was used for enrichment of the sensitivity for detection of mutations, such as the *BRAF* mutation, *NRAS* mutation [24], and *KRAS* mutations [24–26], and even detection of disease-related hypermethylation [27].

To enrich low levels of *NPM1* mutations in this study, the E-ice-COLD-PCR assay was developed for the enrichment of *NPM1* mutations, which provides the strongest enrichment for all types of mutations as compared to the full-, fast-, and ice-COLD-PCR methods. A non-extendable blocker probe is very important to block wild-type amplification, which incorporates with chemically modified nucleotides or locked nucleic acid (LNA) bases into the blocker probe. The addition of LNA into the blocker probe increases the stability and Tm of the complete match DNA-LNA probe heteroduplex, leading to a higher Tm difference between the complete match and mismatch DNA-LNA heteroduplex compared with and without incorporated LNA [28]. Primers for E-ice-COLD-PCR should be designed to a have similar Tm, range 55–60˚C, with the recommended Tm of the blocker probe being 7–10˚C higher than that of the Tm of the primers. Self-complementarity of primers or probes or primer-probes decrease the efficiency of the E-ice-COLD-PCR assay for enrichment of detecting low levels of mutations. In previous studies, the LNA blocker probes were usually designed based on the mutation site, which should be located in the center of the probe and appropriate for detection of point mutations or SNPs. In this study, there were insertion mutations. The LNA blocker probe was designed differently from other studies, with the mutation site located closely to the 3' end of the probe, leading to a higher Tm difference between the complete match and

mismatch DNA-LNA heteroduplex. Because a non-extendable blocker probe of E-ice-COLD PCR blocked wild-type amplification, only mutant PCR product was produced. Consequently, E-ice-COLD PCR could enhance the low levels of mutated gene amplification, increase the sensitivity of post-PCR analysis method, increase the accuracy of result interpretation, and reduce the misdetection of *NPM1* mutations. Consequently, the patients were treated with appropriate treatment.

A total of 83 patient samples were included in this study—nine samples were mutated *NPM1* genes and 74 samples were wild-type as determined by a standard PCR assay followed by 10% polyacrylamide gel electrophoresis, which can be separated by a four bp difference. All products were confirmed by direct sequencing—eight samples had *NPM1* mutation type A, which inserts TCTG, and 75 samples were wild-type. One sample showed inconsistent results between the conventional PCR assay followed by polyacrylamide gel electrophoresis and direct sequencing. Mutations by the conventional PCR assay followed by polyacrylamide gel electrophoresis were verified by direct sequencing as the wild-type. Perhaps, the low level of mutations in the wild-type background interfered with the direct sequencing, leading to verification of the mutation sample as the wild-type. Nine samples contained mutated *NPM1* genes and 74 samples were wild-type as determined by the E-ice-COLD-PCR assay combined with HRM analysis. All PCR products confirmed by direct sequencing showed 100% concordance with the E-ice-COLD-PCR assay. The most common mutation was type A, which inserts TCTG.

The E-ice-COLD-PCR assay combined with HRM analysis enhances the detection of low levels of *NPM1* mutations in a mutant/wild-type mixture compared to that with conventional PCR. This method is important for prognostic information and the determination of appropriate therapeutic interventions in patients with AML. However, this method cannot identify the type of mutation unless the scanning is followed by sequencing of the positive samples due to slight differences in the Tm between each *NPM1* mutation type. Moreover, performing the E-ice-COLD-PCR assay prior to direct sequencing allows for substantially increasing the LOD of the sequencing method. In addition, the E-ice COLD-PCR protocol for *NPM1* mutations requires 5 ng genomic DNA, which is a low DNA concentration for detecting *NPM1* mutations in patients with AML as compared that for conventional PCR, which requires a DNA concentration of at least 100 ng genomic DNA. Several studies have shown that 2–10 ng of DNA was used as a template for the E-ice COLD-PCR assay for detecting the mutations [25–27]. HRM analysis is an inexpensive scanning method and does not require opening the PCR tubes, preventing contamination and, more importantly, shorter turn around time and easier interpretation of the results. Interestingly, the E-ice-COLD-PCR protocol in this study can be performed on a real-time PCR machine followed by HRM analysis or conventional PCR machine followed by agarose gel electrophoresis. However, both methods can also be integrated in the routine molecular diagnosis of AML using standard laboratory equipment. In addition, when compare the cost of E-ice-COLD PCR combined with HRM analysis for detecting *NPM1* mutations per individual patient to routine methods in Thailand including FAST PCR assays with agarose E-gel electrophoresis and standard PCR with polyacrylamide gel electrophoresis found that the E-ice-COLD PCR combined with HRM analysis is cheaper than those of routine methods. Therefore, AML patients could have paid low cost for high quality of molecular diagnosis of AML.

*NPM1* mutations can be considered markers of minimal residual disease (MRD) [15, 22]. The method used in the routine monitoring of low levels of MRD is qPCR, but it requires high concentrations of DNA. Therefore, the E-ice COLD-PCR method is a simple and effective way to enhance the detection of MRD and can be applied in the detection of other makers for monitoring MRD. Moreover, a highly sensitive method such as the E-ice-COLD-PCR assay is

desirable for the detection of rare circulating tumor cells or application for solid tumors, such metastatic lesions, with a low percentage of tumor.

## Conclusions

The E-ice-COLD-PCR assay with HRM provides highly specific, short turn around time and sensitive screening for the detection of *NPM1* mutations with easy interpretation of the results, which is great importance for decision-making regarding the treatments for patients with AML. This method detected *NPM1* mutations covering four types of mutations, which are the most prevalent mutations found in Thailand. The LOD for detecting *NPM1* mutations was 12.5% mutant in the final concentration of 5 ng genomic DNA. Out of 83 samples from patients with AML, nine samples contained *NPM1* mutations that were identified as mutation type A. All mutations were detected by this method. In addition, performing the E-ice-COLD-PCR assay prior to direct sequencing allows for substantially increasing the LOD of direct sequencing. Moreover, the E-ice-COLD-PCR protocol can be easily integrated into the routine molecular diagnosis of AML using standard laboratory equipment, as the E-ice-COLD-PCR assay can use a conventional PCR machine and is visualized by agarose gel electrophoresis for detecting *NPM1* mutations.

## Supporting information

**S1 Fig. The results of detecting *NPM1* gene mutations in total 83 patient samples using a standard PCR assay.** The PCR products are visualized on 10% polyacrylamide gel. MT indicates *NPM1* mutated control and WT indicates *NPM1* wild-type control. A double band indicates a heterogeneous mutation.
(DOCX)

**S2 Fig. Detecting *NPM1* gene mutations by HRM analysis.** Detecting *NPM1* gene mutations using the E-ice-COLD-PCR assay and generating the data into HRM difference curves. Red line indicates mutant control. Green line indicates wild-type control. The wild-type samples present the peak at the same location as the wild-type control, and the mutant samples present the peak at the same location as the mutant control.
(DOCX)

**S1 Data.**
(DOCX)

**S1 Raw images.**
(PDF)

## Acknowledgments

The authors would like to thank the Faculty of Associated Medical Sciences and Faculty of Medicine, Chiang Mai University for the research facility support. We would also like to thank all colleagues of the Division of Hematology that contributed to blood sample collection.

## Author Contributions

**Conceptualization:** Suwit Duangmano.

**Formal analysis:** Rattana Kongta.

**Funding acquisition:** Rattana Kongta, Suwit Duangmano.

**Investigation:** Rattana Kongta.

**Methodology:** Rattana Kongta.

**Project administration:** Suwit Duangmano.

**Resources:** Noppamas Panyasit, Wuttichote Jansaento.

**Supervision:** Suwit Duangmano.

**Validation:** Suwit Duangmano.

**Writing – original draft:** Rattana Kongta.

**Writing – review & editing:** Suwit Duangmano.

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
