## [Decision Letter · Decision Letter 0]

23 Jun 2022

PONE-D-22-05079Development of E-ice-COLD-PCR assay combined with HRM analysis for Nucleophosmin1 gene mutation detection in acute myelogenous leukemiaPLOS ONE

Dear Dr. Duangmano,

Thank you for submitting your manuscript to PLOS ONE. After careful consideration, we feel that it has merit but does not fully meet PLOS ONE’s publication criteria as it currently stands. Therefore, we invite you to submit a revised version of the manuscript that addresses the points raised during the review process.

We look forward to receiving your revised manuscript.

Kind regards,

Daniel Thomas, MD

Academic Editor

PLOS ONE

Journal Requirements:

In your cover letter, please note whether your blot/gel image data are in Supporting Information or posted at a public data repository, provide the repository URL if relevant, and provide specific details as to which raw blot/gel images, if any, are not available. Email us at plosone@plos.org if you have any questions

Additional Editor Comments (if provided):

Good method development paper; important test for patient management, any improvements in sensitivity and turn around time celebrated. Will likely be citied.

Minor points:

Check/provide reference that AML is responsible for most number of deaths from blood cancer in Thailand

Abstract: define HRM, reduce number of abbreviations if possible

Line 61: poor grammar: results of NPM1 molecular analysis change management

Turn around time is perhaps better english than time to result

Can the authors provide a discussion of similar methods for other molecular markers/ other cancers in discussion? and brief discussion on cost vs routine methods

Discussion: Could emphasis that VAF of NPM1 is also important for prognosis and its order of acquisition: See Figure 4f Nat Commun

. 2021 Dec 13;12(1):7244. doi: 10.1038/s41467-021-27472-5.

Reviewers' comments:

Reviewer's Responses to Questions

**Comments to the Author**

1. Is the manuscript technically sound, and do the data support the conclusions?

Reviewer #1: Partly

Reviewer #2: Yes

2. Has the statistical analysis been performed appropriately and rigorously? 

Reviewer #1: No

Reviewer #2: No

3. Have the authors made all data underlying the findings in their manuscript fully available?

Reviewer #1: Yes

Reviewer #2: Yes

4. Is the manuscript presented in an intelligible fashion and written in standard English?

Reviewer #1: No

Reviewer #2: No

5. Review Comments to the Author

Reviewer #1: The authors have attempted to develop and report a method to characterise Nucleophosmin1 gene mutation in AML samples using E-ice-COLD-PCR assay combined with HRM analysis. The study cohort used include 83 samples and they have detected NPM1 mutations in 9 patients with AML using the E-ice-COLD-PCR/HRM analysis assay. The sincere efforts and hard work of the authors in collecting the samples, developing the assay, optimising it and compiling the data are evident in the presented study. However, the manuscript in its current form is not suitable for publication as the presentation of the manuscript does little justice to the presented work.

The results, figures and discussion are presented poorly to fully evaluate the findings reported. For instance, in section, Detecting NPM1 gene mutations in patient samples using the E-ice-COLD-PCR assay, opening sentence mentions "NPM1 mutations were detected in a total of 83 patient samples using the E-ice-COLD-PCR assay" which is wrong. Several inaccurate and misleading sentences are present throughout the manuscript. Similarly, the figures and associated legends should be revised as it is difficult to evaluate them in their current form. For example, In Figure 1, the authors could indicate which samples they believe to include double bands as wild type in gel that includes samples 60 to 65 appears to have have a double band and the contrasts are different between gel pictures.

It is very difficult to understand clearly and appreciate why and how the E-ice-COLD-PCR assay is better than the other existing assays. In the current manuscript, a one line limitation statement ("However, most of these methods are technically challenging, complicated, and expensive") is provided and it would be beneficial to have a clear comparison with an existing method and highlight how E-ice-COLD-PCR assay is more suitable showing better accuracy.

The data shows 9 positive cases and one that has discrepancy between the sequencing and the PCR assay. However, the results sections do not describe the data well. The results include figure legends instead of clear description of the data. Thus, the presentation of the results section makes it very hard to evaluate the figures, data in general and the methods. Therefore, major, rigorous revision of all the sections and complete rewriting of the results section with clear introduction that places the use of NPM1 mutations detection in resource limited clinical settings including Thailand is needed before this can be considered for Publication. The discussion can be revised to actually focus on the data presented and how this compares to other currently available methods and the significance of the assay.

The presented study can be useful and may have a potential to contribute towards AML stratification and MRD but the manuscript requires major revision.

Reviewer #2: In general, the work is well written, aiming to develop a quick and cost effective method for the detection of NPM1 mutation in acute myeloid leukemia patients. However, I believe that some important points need to be answered by the authors before the work can be considered for publication. My comments are as follows:

1. Abstract, line 25: I think the authors need to use the word "ability" instead of "inability", since it refers to the limitation of the previous detection methods.

2. Abstract, line 26: "HRM" is written for the first time, thus, must be written in full.

3. Abstract, line 27-28: Is the samples from AML patients with normal karyotype?

4. Introduction: I suggest the authors carefully check each of the statement that need to cite the reference. For examples, "AML is the major cause of death among hematological malignancies in Thailand" and "Recently, researchers have suggested that mutations of the nucleophosmin 1 (NPM1) gene represent the most frequent molecular alterations in AML that affect the processes of cellular differentiation and apoptosis, especially in the presence of a normal karyotype." There are other few statements.

5. Result, line 222-223: I suggest the authors write or label the respective patient samples with the mutation.

6. Result, line 239-241: Are you referring to Fig. 2a?

7. Result, line 289-290: The authors state that "The results showed that nine samples contained NPM1 mutations, and the remaining samples were wild-type" in reference to Fig. 5. However, the number of mutant curve in Fig. 5 does not correspond with the number of samples with NPM1 mutation. Please clarify.

8. Result, line 295: I suggest the authors to show the calculation of PPV and NPV.

9. Is the manuscript has been proofread? If not, I suggest that the authors get editing help from someone with full professional proficiency in English.

6. PLOS authors have the option to publish the peer review history of their article (what does this mean?). If published, this will include your full peer review and any attached files.

Reviewer #1: No

Reviewer #2: No

---

## [Author Response · Author response to Decision Letter 0]

18 Aug 2022

1. Additional Editor Comments:

1.1 Check/provide reference that AML is responsible for most number of deaths from blood cancer in Thailand

Answer: We have already revised the sentence 

1.2 Abstract: define HRM, reduce number of abbreviations if possible

Answer: We have revised

1.3 Line 61: poor grammar: results of NPM1 molecular analysis change management

Answer: We didn’t find this sentence at line 61

1.4 Turn around time is perhaps better english than time to result

Answer: We have revised

1.5 Can the authors provide a discussion of similar methods for other molecular markers/ other cancers in discussion? and brief discussion on cost vs routine methods

Answer: We have provided a discussion at line 361-383 and 436-441 

2. Reviewer #1:

2.1 The results, figures and discussion are presented poorly to fully evaluate the findings reported. For instance, in section, Detecting NPM1 gene mutations in patient samples using the E-ice-COLD-PCR assay, opening sentence mentions "NPM1 mutations were detected in a total of 83 patient samples using the E-ice-COLD-PCR assay" which is wrong. Several inaccurate and misleading sentences are present throughout the manuscript. Similarly, the figures and associated legends should be revised as it is difficult to evaluate them in their current form. For example, In Figure 1, the authors could indicate which samples they believe to include double bands as wild type in gel that includes samples 60 to 65 appears to have have a double band and the contrasts are different between gel pictures.

Answer: We have revised the sentence that represent inaccurate or misleading sentence. The contrasts are difference between gel because we analyzed PCR products by acrylamide gel electrophoresis for 5-6 samples per a gel, but we have been tried to adjust the contrast for each gel in similar condition. However, the results are still difficult interpretation. This is a reason that why we develop new method for detecting NPM1 mutations.

2.2 It is very difficult to understand clearly and appreciate why and how the E-ice-COLD-PCR assay is better than the other existing assays. In the current manuscript, a one line limitation statement ("However, most of these methods are technically challenging, complicated, and expensive") is provided and it would be beneficial to have a clear comparison with an existing method and highlight how E-ice-COLD-PCR assay is more suitable showing better accuracy.

Answer: We have provided the discussion at line 367-378 and 401-405

2.3 The data shows 9 positive cases and one that has discrepancy between the sequencing and the PCR assay. However, the results sections do not describe the data well. The results include figure legends instead of clear description of the data. Thus, the presentation of the results section makes it very hard to evaluate the figures, data in general and the methods. Therefore, major, rigorous revision of all the sections and complete rewriting of the results section with clear introduction that places the use of NPM1 mutations detection in resource limited clinical settings including Thailand is needed before this can be considered for Publication. The discussion can be revised to actually focus on the data presented and how this compares to other currently available methods and the significance of the assay.

The presented study can be useful and may have a potential to contribute towards AML stratification and MRD but the manuscript requires major revision.

 Answer: We have revised 

3. Reviewer #2

3.1 Abstract, line 25: I think the authors need to use the word "ability" instead of "inability", since it refers to the limitation of the previous detection methods.

Answer: We have revised at line 26

3.2 Abstract, line 26: "HRM" is written for the first time, thus, must be written in full.

Answer: We have revised.

3.3 Abstract, line 27-28: Is the samples from AML patients with normal karyotype?

 Answer: The samples weren’t examined the karyotype test because in this study we aim to develop the highly specific and sensitive screening method for NPM1 mutation in AML patients. We don’t concern about the karyotype of AML samples using validate the developed method. So, we use all types of AML samples without the identification of normal or abnormal karyotype to validate the developed method.

3.4 Introduction: I suggest the authors carefully check each of the statement that need to cite the reference. For examples, "AML is the major cause of death among hematological malignancies in Thailand" and "Recently, researchers have suggested that mutations of the nucleophosmin 1 (NPM1) gene represent the most frequent molecular alterations in AML that affect the processes of cellular differentiation and apoptosis, especially in the presence of a normal karyotype." There are other few statements.

Answer: We have already revised the sentence

3.5 Result, line 222-223: I suggest the authors write or label the respective patient samples with the mutation.

Answer: We have revised at line 240-241.

3.6 Result, line 239-241: Are you referring to Fig. 2a?

Answer: Yes, we are. We have revised.

3.7 Result, line 289-290: The authors state that "The results showed that nine samples contained NPM1 mutations, and the remaining samples were wild-type" in reference to Fig. 5. However, the number of mutant curve in Fig. 5 does not correspond with the number of samples with NPM1 mutation. Please clarify.

Answer: The fig 5, we represented the HRM difference curve from few mutant and wild-type samples because we would like to give some examples of data generated from HRM analysis. In addition, we represented mutant and wild-type results after HRM analysis in the same figure because we would like to show the differences between mutant and wild-type HRM difference curve in the same figure. For HRM analysis of total 83 patient samples, the PCR products were analyzed by HRM analysis, and the data generated in separated HRM difference curve from each other. The separation of HRM analysis in each sample resulting in clearly interpretation of HRM difference curve of patient samples. We show the results of the total 83 patients in supplementary data.

3.8 Result, line 295: I suggest the authors to show the calculation of PPV and NPV.

Answer: We have described the calculation of PPV and NPV at line 222-227.

3.9 Is the manuscript has been proofread? If not, I suggest that the authors get editing help from someone with full professional proficiency in English.

Answer: Yes, it is. We have proofread the manuscript.

---

## [Editor Report · Decision Letter 1]

22 Aug 2022

Development of E-ice-COLD-PCR assay combined with HRM analysis for Nucleophosmin1 gene mutation detection in acute myelogenous leukemia

PONE-D-22-05079R1

Dear Dr. Duangmano,

We’re pleased to inform you that your manuscript has been judged scientifically suitable for publication and will be formally accepted for publication once it meets all outstanding technical requirements.

Kind regards,

Daniel Thomas, MD

Academic Editor

PLOS ONE

Additional Editor Comments (optional):

Editor is satisfied that all reviewers' comments have been adequately addressed.
---

## [Editor Report · Acceptance letter]

5 Sep 2022

PONE-D-22-05079R1 

Development of E-ice-COLD-PCR assay combined with HRM analysis for Nucleophosmin1 gene mutation detection in acute myelogenous leukemia 

Dear Dr. Duangmano:

I'm pleased to inform you that your manuscript has been deemed suitable for publication in PLOS ONE. Congratulations! Your manuscript is now with our production department. 

Kind regards, 

on behalf of

Dr. Daniel Thomas 

Academic Editor

PLOS ONE